# Mechanism of Action of Inhaled Insulin on Whole Body Glucose Metabolism in Subjects with Type 2 Diabetes Mellitus

**DOI:** 10.3390/ijms20174230

**Published:** 2019-08-29

**Authors:** Rucha J. Mehta, Amalia Gastaldelli, Bogdana Balas, Andrea Ricotti, Ralph A. DeFronzo, Devjit Tripathy

**Affiliations:** 1Diabetes Division, Department of Medicine, University of Texas Health Science Center, San Antonio, TX 78229, USA; 2Cardiometabolic Risk Laboratory, Institute of Clinical Physiology, 56124 Pisa, Italy; 3South Texas Veteran Health Care System, San Antonio, TX 78229, USA

**Keywords:** inhaled insulin, whole body glucose metabolism, hepatic glucose production, tracers, OGTT, disposition index

## Abstract

In the current study we investigate the mechanisms of action of short acting inhaled insulin Exubera®, on hepatic glucose production (HGP), plasma glucose and free fatty acid (FFA) concentrations. 11 T2D (Type 2 Diabetes) subjects (age = 53 ± 3 years) were studied at baseline (BAS) and after 16-weeks of Exubera® treatment. At BAS and after 16-weeks subjects received: measurement of HGP (3-^3^H-glucose); oral glucose tolerance test (OGTT); and a 24-h plasma glucose (24-h PG) profile. At end of study (EOS) we observed a significant decrease in fasting plasma glucose (FPG, 215 ± 15 to 137 ± 11 mg/dl), 2-hour plasma glucose (2-h PG, 309 ± 9 to 264 ± 11 mg/dl), glycated hemoglobin (HbA1c, 10.3 ± 0.5% to 7.5 ± 0.3%,), mean 24-h PG profile (212 ± 17 to 141 ± 8 mg/dl), FFA fasting (665 ± 106 to 479 ± 61 μM), post-OGTT (433 ± 83 to 239 ± 28 μM), and triglyceride (213 ± 39 to 120 ± 14 mg/dl), while high density cholesterol (HDL-C) increased (35 ± 3 to 47 ± 9 mg/dl). The basal HGP decreased significantly and the insulin secretion/insulin resistance (disposition) index increased significantly. There were no episodes of hypoglycemia and no change in pulmonary function at EOS. After 16-weeks of inhaled insulin Exubera® we observed a marked improvement in glycemic control by decreasing HGP and 24-h PG profile, and decreased FFA and triglyceride concentrations.

## 1. Introduction

Type 2 diabetic patients are characterized by fasting and postprandial hyperglycemia [1]. Fasting hyperglycemia primarily results from an increase in basal endogenous, primarily hepatic, glucose production (HGP) due to an increase in hepatic gluconeogenesis [1,2,3,4,5,6]. Postprandial hyperglycemia results from insulin resistance in muscle [7], impaired suppression of hepatic glucose production [3,8], and decreased insulin secretion [9,10]. Clinical trials have demonstrated that inhaled insulin, administered thrice daily with meals, in addition to controlling post-prandial glucose levels also causes normalization/near-normalization of the fasting plasma glucose (FPG) concentration in patients with type 2 diabetes (T2D) who are inadequately controlled on oral agent therapy [11]. Although Exubera® was initially approved in 2006 by Food and Drug Administration (FDA) [12], it was withdrawn from the US in 2007 for poor patient acceptance and thus lack of commercial viability. However, another inhaled insulin Afrezza®, which is similar to Exubera®, has been approved by FDA and is currently available [13]. There is renewed interest in alternative routes of insulin delivery in order to improve patient adherence to therapy, as well as improve quality of life. This novel delivery route may also combat physician and patient inertia in intensifying treatment when needed. 

The major determinant of the FPG is the basal rate of HGP that prevails throughout the sleeping hours [1,2,5]. Because Exubera® is a short acting (4–6 hours) inhaled insulin, it is somewhat paradoxical that three daily administrations (7–8 a.m., 12–1 p.m., 6–7 p.m.) could influence the basal rate of HGP that prevails from 12 a.m. to 7–8 a.m. on the following day. This has important clinical implications since it implies that a ‘short acting’ insulin preparation can normalize/cause near-normalization of the FPG concentration, obviating the need for a ‘long acting’ basal insulin.

In patients with Type 1 Diabetes Mellitus (T1D) and T2D, the serum insulin concentration reaches its peak faster after inhalation of EXUBERA® (49 min, range = 30–90 min) than after subcutaneous injection of human regular insulin (105 min, range = 60–240 min) [12]

No previous study has examined in humans the effect of inhaled insulin on the overnight basal rate of HGP or the mechanisms via which inhaled insulin inhibits basal HGP and reduces the FPG concentration. Potential mechanisms include: (i) inhibition of gluconeogenesis secondary to reduced lactate flux through the Cori cycle and/or decreased glycerol flux due to inhibition of lipolysis, (ii) reversal of lipotoxicity [1,14], and (iii) reversal of glucotoxicity [1,15,16]. Thus, augmented glucose uptake by inhaled insulin following each meal reduces postprandial hyperglycemia, leading to a reduction in the mean day-long glucose level with reversal of hepatic glucotoxicity. Enhanced muscle glucose uptake by Exubera® and more effective conversion of glucose to glycogen, as well as enhanced muscle glucose oxidation, would be expected to reduce muscle efflux of lactate, thereby decreasing substrate availability for gluconeogenesis.

In the present study we have, for the first time, examined the effect of inhaled insulin on HGP in poorly controlled (defined as glycated hemoglobin HbA1c≥8 %) patients with T2D. These results have important implications for the mechanisms of action of inhaled insulin on whole body glucose metabolism in patients with T2D.

## 2. Results

### 2.1. Glycemic Control

Following 16 weeks of inhaled insulin therapy, fasting (215 ± 15 to 137 ± 11 mg/dl, *p* < 0.0002) and 2-h plasma glucose (309 ± 9 to 264 ± 11 mg/dl, *p* < 0.03) concentration and glucose AUC during OGTT (35,575 ± 1159 to 28586 ± 1630 mg/dl x 120 min, *p* < 0.004, Figure 1A) decreased significantly. HbA1c decreased from 10.3 ± 0.5 to 7.5 ± 0.3%, *p* < 0.0001. 

The mean plasma glucose concentration during the 24-h plasma glucose profile (212 ± 17 to 141 ± 8 mg/dl Figure 2A) was markedly decreased (*p* < 0.0001) following Inhaled insulin treatment, with a greater decline in the postprandial compared to fasting glucose.

### 2.2. Body Weight

Following 16 weeks of treatment with inhaled insulin, body weight increased about 2 kg (*p* = 0.01), Table 1.

### 2.3. Plasma Lipids

Fasting (665 ± 106 to 479 ± 61 μM, *p* = 0.05) and post-OGTT (mean 0-120, 433 ± 83 to 239 ± 28 μM, *p* = 0.02, Figure 1C) plasma FFA concentration decreased significantly following treatment with inhaled insulin. 

The decline in fasting plasma FFA concentration did not correlate with the decline in HGP or with the improvement in Matsuda index of insulin sensitivity or with the increase in beta cell function. Plasma triglyceride decreased, while plasma HDL increased following inhaled insulin (Table 1). LDL and total cholesterol did not change significantly. 

### 2.4. Hepatic Glucose Production (HGP)

The basal rate of HGP before treatment (11.5 ± 0.9 µmol/kg.min) was significantly higher than in historical NGT control subjects (9.8 ± 1.2 µmol/kg.min, [17]) and decreased significantly after 16 weeks of inhaled insulin (to 8.9 ± 0.5 µmol/kg.min, *p* = 0.017, Figure 3). The decline in basal HGP correlated with the decline in FPG and weakly with the decline in HbA1c (Table 2).

### 2.5. Lactate Turnover

The fasting plasma lactate concentration did not change significantly following inhaled insulin (1550 ± 155 to 1466 ± 168 µM). The basal rate of lactate turnover also did not change significantly (4.54 ± 0.64 to 4.57 ± 0.55 μmol/kg∙min). Neither the percentage of glucose derived from lactate (19.7 ± 2.0 to 20.9 ± 3.9 %) nor the rate of gluconeogenesis from lactate (2.23 ± 0.3 to 2.01 ± 0.44 μmol/kg∙min) changed significantly following inhaled insulin.

### 2.6. Insulin Secretion and Insulin Sensitivity

Following 16 weeks of Inhaled insulin, the mean incremental plasma C-peptide concentration during the 0–30 min (28.5 ± 9.9 vs 36.4 ± 9.8 ng/ml) and 0–120 minute (307 ± 63 vs 407 ± 78 ng/ml) time period of the OGTT did not change (*p* = ns). The fasting plasma C-peptide (CP) concentration (5.5 ± 0.9 vs. 4.2 ± 0.9 ng/ml) also was unchanged following inhaled insulin treatment. The insulinogenic index 0–30 min (ΔCP_0-30_/ΔG_0-30_) and the 2-h response of C-peptide factored by incremental glucose(ΔCP_0-120_/ΔG_0-120_) did not change significantly (Table 3). The Matsuda index of insulin sensitivity almost doubled following inhaled insulin (*p* < 0.05). Consequently, the insulin secretion/insulin resistance (disposition) index of beta cell function during the 0–30 min and 0–120 min time periods increased significantly (both *p* < 0.02) (Table 3). As expected, the mean 24-hr endogenous insulin secretion (Figure 2D) was lower following 4 months of inhaled insulin therapy. 

The improvement in Matsuda index of insulin sensitivity correlated with the decline in FPG (r = 0.65, *p* < 0.05) but not with the decline in HbA1c. The improvement in beta cell function tended to correlate with both the decline in FPG and HbA1c (see Table 2). The number of subjects in the present study was relatively small and more robust correlations may have been seen with a larger sample size.

### 2.7. Plasma Glucagon

The fasting plasma glucagon and mean 24-h plasma glucagon (Table 3) concentrations did not change significantly following inhaled insulin. 

### 2.8. Adverse Events Monitoring

There were no episodes of hypoglycemia and there was no significant change in FEV1 following treatment with inhaled insulin (100 ± 5 vs. 96 ± 7 % of predicted). No other drug-related side effects were reported by any subject. One subject had worsening of his gout, but this was felt to be unrelated to inhaled insulin treatment.

## 3. Discussion

In our study short acting inhaled insulin Exubera® decreased HbA1c, fasting and post prandial blood glucose significantly in Type 2 diabetic patients over a period of 16 weeks. The decline in FBS correlated with the decline in the HGP. Additionally mean 24-h PG profile, fasting and post OGTT FFA, triglyceride decreased, while high density cholesterol (HDL-C) increased. The insulin secretion/insulin resistance (disposition) index increased significantly. There were no episodes of hypoglycemia and no change in pulmonary function following Exubera®.

Both type 1 and type 2 diabetic patients use insulin for the treatment of hyperglycemia. Insulin injection therapy is difficult and painful for many patients, thus newer routes for insulin administration are current direction of insulin research, including administration through inhalation. The first approved inhaled insulin (Exubera®, in 2006) was withdrawn in 2007, mainly because of poor sale, but after the commercialization of Afrezza® [13] there is new interest in inhaled insulin [18]. 

Although inhaled insulin is used before each meal and does not substitute long acting insulin it has been shown to decrease not only postprandial, but also fasting, hyperglycemia. However, the mechanisms are still not well understood. In this study we used tracer techniques and continuous glucose monitoring to evaluate changes in glucose fluxes and insulin resistance and understand the mechanisms via which inhaled insulin improves glucose homeostasis in patients with T2D. 

Our results demonstrate that 16 weeks of inhaled insulin treatment caused a marked improvement in global glycemic control, as evidenced by decreases in glucose concentrations after overnight fasting, during OGTT (both 2-h and mean glucose), HbA1c, and mean 24-h plasma glucose profile. Administration of inhaled insulin thrice daily markedly reduced the HbA1c by 2.8% from 10.3 to 7.5% and the mean 24-h plasma glucose concentration by 33%, from 212 to 141 mg/dl. The improvement in day-long hyperglycemia was accounted for by a reduction in FPG glucose by 74 mg/dl and mean postprandial plasma glucose concentration by 61 mg/dl (24-h glucose monitoring). A similar reduction in mean postprandial glucose concentration (58 ± 15 mg/dl) was observed during the OGTT. Because of earlier peak and faster clearance, inhaled insulin has been shown to reduce the risks of hypoglycemia and also able to achieve comparable glycemic control with basal insulin [19,20]. In the current study we did not include a control group as it was primarily designed to explain mechanisms whereby a short acting inhaled insulin can also reduce fasting plasma glucose. Diet and exercise can influence whole body glucose homeostasis; hence subjects were asked to maintain a similar weight maintaining diet and exercise regimen throughout the study duration. No weight loss was observed in any of the subjects and in fact there was a mean weight gain in the group. Hence, we believe that diet and exercise likely do not contribute to any of the glycemic benefits we see at beginning and end of study.

Since the primary determinant of the FPG concentration is the rate of endogenous glucose production primarily HGP [1,2,3,4,6], we used tracer infusion to measure in vivo glucose fluxes. Inhaled insulin reduced the basal rate of HGP by 23% from 11.5 to 8.9 μmol/kg.min and the decrement in HGP was correlated with the decrement in FPG concentration (r = 0.66, *p* = 0.05) and weakly with the decline in HbA1c. One outlier accounted for the failure to observe a significant correlation with the HbA1c. Since the fasting plasma insulin (FPI) concentration did not change and was, in fact, slightly reduced, hyperinsulinemia cannot explain the reduction in basal HGP. Inhaled insulin therapy had no effect on the fasting plasma glucagon concentration and there was no correlation between the decrement in HGP and plasma glucagon concentration following Inhaled insulin, excluding inhibition of glucagon secretion as a cause of the decline in HGP [21,22].

Multiple studies [1,3,5,6] have documented that an accelerated rate of gluconeogenesis is the primary cause of the increase in basal HGP. In poorly controlled T2D individuals with fasting hyperglycemia, a significant amount of the glucose that is taken up by muscle (secondary to the mass action effect of hyperglycemia) enters the glycolytic cycle but cannot be oxidized and leaves the cell as lactate [23]. In the liver the lactate is taken up and converted to glucose, i.e., the Cori cycle [17]. In the present study, inhaled insulin failed to alter the basal rate of lactate turnover or gluconeogenesis derived from lactate excluding reduced lactate-derived gluconeogenesis as a cause of the reduction in basal rate of HGP. Hyperinsulinemia also inhibits lipolysis in adipocytes, as evidenced by the decrease in fasting and post-OGTT plasma FFA concentration in the present study. The decline in plasma FFA concentration would be expected to inhibit PEPCK, the rate limiting enzyme for gluconeogenesis [24]. However, we failed to observe a significant correlation between the decline in plasma FFA concentration and decline in HGP. Plasma glycerol concentration was not measured in the present study, but is likely to have declined in concert with the decrease in plasma FFA concentration. A decline in plasma glycerol concentration following inhibition of lipolysis by insulin would be expected to reduce glycerol-gluconeogenesis.

Lastly, chronically elevated plasma glucose levels upregulate hepatic glucose-6-phosphatase activity, the rate limiting enzyme for hepatic glucose release [14]. Reduction in the mean plasma glucose concentration by inhaled insulin from 8:00 a.m. to 12:00 a.m. may be sufficient to reverse this glucotoxic effect on the liver and down regulate glucose-6-phosphatase throughout the sleeping hours (12:00 a.m. – 8:00 a.m.), leading to a reduction in basal HGP and FPG concentration. Consistent with a role for glucotoxicity in the accelerated rate of HGP, the reduction in FPG concentration was strongly correlated with the decrease in basal rate of HGP, which occurred despite unchanged fasting plasma insulin and glucagon concentrations. Multiple factors, including inhibition of gluconeogenesis due to decreased substrate (glycerol) availability and reversal of glucotoxicity, are likely to contribute to the decline in basal HGP. 

Four months of inhaled insulin therapy significantly enhanced insulin sensitivity (Matsuda index) and beta cell function (insulin secretion/insulin resistance index). The increase in insulin sensitivity was correlated with the improvement in FPG, while the improvement in beta cell function tended to be correlated with the decrement in both FPG and HbA1c. Although correlations do not prove causality, they are consistent with reversal of the glucotoxic effect of chronic hyperglycemia on tissue sensitivity to insulin and insulin secretion [1,15,16,21]. Although FFA concentrations declined following inhaled insulin therapy, we failed to observe any correlation between the decrease in plasma FFA and the improvements in insulin sensitivity or beta cell function, making reversal of lipotoxicity [1,14] a less likely explanation for the enhanced insulin sensitivity and beta cell function. The results of this study are in agreement with previous data in animal models. Edgerton et al. administered to dogs human insulin via inhalation (Exubera®; *n* = 9) or infusion (Humulin R; *n* = 9) using an infusion algorithm that yielded matched plasma insulin kinetics between the two groups to determine whether this insulin effect lasts for a prolonged duration such that it could explain the effect observed in diabetic patients [25]. Somatostatin was infused to prevent insulin secretion, and glucagon was infused to replace basal plasma levels of the hormone. Glucose was infused into the portal and peripheral vein achieving virtually identical arterial and hepatic sinusoidal insulin and glucose levels in the two groups. Notwithstanding, glucose utilization was greater when insulin was administered by inhalation that caused a greater increase in non-hepatic glucose uptake in the first 3 h after inhalation; thereafter, net hepatic glucose uptake was greater. Inhalation of insulin was associated with greater than expected (based on insulin levels) glucose disposal. 

Another inhaled insulin was recently developed by Technosphere®. Rave and colleagues assessed the time action profile and within- and between-subject variability of inhaled Technosphere® Insulin (TI) compared with subcutaneous regular human insulin (s.c. RHI) in 13 human subjects [26]. Technosphere Insulin had a more rapid onset of action than s.c. RHI, showing around 60% of the glucose-lowering effect during the first 3 hours after application vs less than 30% obtained with s.c. RHI. TI is more rapidly absorbed than subcutaneous insulin therapies, has a shorter duration of action and is associated with less hypoglycemia. 

Overall, poorly controlled T2D patients are characterized by a state of decompensated metabolic control. Institution of inhaled insulin therapy, by improving metabolic control and ameliorating glucotoxicity, and possibly lipotoxicity, can lead to enhanced insulin sensitivity and beta cell function and may serve as an alternative insulin agent in patients reluctant to administer multiple subcutaneous injections of insulin daily or in patients who experience late postprandial hypoglycemia with subcutaneous insulin.

The present study has several limitations, i.e., the relatively small number of subjects and the lack of a group of subjects treated with conventional insulin therapy for comparison with inhaled insulin, as it was designed to understand the mechanisms of action of inhaled insulin in improving metabolic control in type 2 diabetic patients already demonstrated in previous studies [19,20,27]. 

In summary, our study shows that thrice daily inhaled insulin administration effectively reduced the fasting and postprandial plasma glucose concentrations and markedly improved glycemic control (ΔHbA1c = −2.8%) in poorly controlled T2D patients by augmenting tissue sensitivity to insulin and enhancing beta cell function. Inhaled insulin may be an advantage in poorly controlled diabetic patients who are averse to taking insulin injections, with faster onset of action and a shorter duration, thereby reducing risk of hypoglycemia. Moreover, due to its beneficial effects on whole body glucose homeostasis, inhaled insulin may prove to be an easier option and may help reduce clinical inertia for the insulin requiring type 2 diabetic patients.

## 4. Subjects and Methods

This was an open-labeled study in which subjects received inhaled insulin Exubera® in addition to their pre-existing regimen of metformin and/or sulfonylurea.

### 4.1. Subjects

11 poorly controlled (HbA1c = 10.3 ± 0.5%; FPG = 215 ± 15 mg/dl) T2D patients (age = 53 ± 3 y; BMI = 33.1 ± 1.4 kg/m^2^; 2 males/9 females; diabetes duration = 4.4 ± 2.5 y) participated in the study. Subjects were required to have a HbA1c ≥ 8.0% and stable body weight (±1.4 kg) for at least 6 months prior to study. Diabetic patients were taking a stable dose of metformin alone (n = 4) or metformin plus sulfonylurea (n = 7) for at least 4 months prior to study. Oral antidiabetic medica tion was not changed during the study.

Patients were excluded if they reported current or prior insulin therapy for > 1 week in the year preceding the study, smoking within 6 months of screening, poorly controlled asthma, clinically significant chronic obstructive pulmonary disease (COPD), or had abnormal pulmonary function as defined by a FEV1 (Forced Expiratory Volume in 1 second) <70% of predicted. All subjects had normal liver, cardiopulmonary, and kidney function as determined by medical history, physical examination, screening blood tests, electrocardiogram, and urinalysis. 

The study protocol was approved by the Institutional Review Board of the University of Texas Health Science Center, San Antonio, (HSC2007025H, approved 10/10/2006) and informed written consent was obtained from all subjects before participation. All studies were performed at the Bartter Research Unit (BRU), Audie L. Murphy Veterans Administration (ALM VA) Hospital, San Antonio at 8:00 a.m. following a 10–12-h overnight fast.

All procedures followed were in accordance with the ethical standards of the responsible committee on human experimentation and with the Helsinki Declaration of 1975, as revised in 2008.

### 4.2. Experimental Methods

#### 4.2.1. OGTT

All patients received a 75-gram oral glucose tolerance test with measurement of plasma glucose, insulin, C-peptide, and FFA concentrations at −30, −15, 0, 15, 30, 45, 60, 75, 90, 105, 120 min. On the day of the OGTT lean body mass was measured with dual energy X-ray absorptiometry (DXA).

#### 4.2.2. Hepatic Glucose Production/Lactate Turnover

Within 2–14 days after the OGTT, subjects returned to the BRU for measurement of hepatic glucose production and lactate turnover following an overnight fast. Subjects consumed their last meal between 6–7 a.m. on the night before study. At 8:00 a.m. on the following day catheters were placed into an antecubital vein for the infusion of all test substances and retrogradely into a vein on the dorsum of the hand for blood withdrawal. The hand was placed in a heated box (70 °C) to obtain arterialized blood.

At 8:00 a.m. blood was collected for basal measurements and prime-continuous infusions of 3-^3^H-glucose (prime = 25 μCi x FPG/90; continuous infusion = 0.25 μCi/min) (DuPont NEN Life Science Products, Boston, MA) and 3-^14^C-lactate (prime = 20 μCi; continuous infusion = 0.2 μCi/min) were started and continued for 3.5 hours. Plasma samples for 3-^3^H-glucose, ^14^C-lactate, and ^14^C-glucose radioactivity and plasma glucose, FFA, lactate, insulin, and C-peptide concentrations [22] were obtained at baseline and at 120, 150, 160, 170, 175, and 180 min after the start of the isotope infusions.

#### 4.2.3. 24-h Glucose/Metabolic Profile

Within 2–10 days after completion of the tracer turnover study, subjects returned to the BRU at 7:30 AM following a 10–12 h overnight fast for a 24-h glucose/metabolic profile. A catheter was inserted into an antecubital vein for all blood withdrawal and a glucose sensor (continuous glucose monitoring system (CGMS), Medtronic MiniMed Inc, Northridge, CA) was placed into the subcutaneous tissue. Following this, subjects met with a dietician who prepared a standardized weight-maintaining breakfast (8:00 a.m.), lunch (12:30 p.m.), and dinner (6:00 p.m.). The diet was comprised of 50% carbohydrate, 30% fat, and 20% protein with caloric distribution as follows: breakfast = 1/5, lunch = 2/5, dinner = 2/5. For 30 min before and 240 min after each meal, blood samples were obtained every 15–30 min for measurement of plasma glucose, insulin, C-peptide, glucagon, and FFA. At 8:00 a.m. on the following morning the antecubital vein catheter and glucose sensor were removed and subjects were allowed to leave.

#### 4.2.4. Inhaled Insulin Therapy

Following the 24-h glucose profile, subjects received an inhaler device and were instructed in its use with empty blisters. Subjects also received a glucose meter (Accucheck, Advantage, Roche Diagnostic, Basel Switzerland) and were instructed in its use. Exubera® consists of blister packets containing 1 mg or 3 mg of human insulin inhalation powder, which are administered using the EXUBERA® inhaler. After an EXUBERA® blister is inserted into the inhaler, the patient pumps the handle of the inhaler and presses a button, causing the blister to be pierced. The insulin inhalation powder is then dispersed into the chamber, allowing the patient to inhale the aerosolized powder. The 1 mg blister packet is equal to ~3 units of subcutaneously injected insulin and the 3 mg blister packet is equal to ~8 units. 

Inhaled insulin was administered 10 min before breakfast, lunch, and dinner and home blood glucose measurements were performed before each meal and at bed time. Each diabetic patient was given an individualized starting insulin dose for each of the three meals based on his/her weight, meal size, time of day, and recent or anticipated exercise. The dose of inhaled insulin was adjusted on a weekly basis after telephone conversation with the physician/diabetes nurse educator or during routine follow up visits. The goals of therapy were to maintain the preprandial glucose concentration between 80–110 mg/dl and the postprandial glucose between 100–150 mg/dl. 

Over the subsequent 4 months, subjects continued to consume a weight maintaining diet containing 50% carbohydrate, 30% fat, and 20% protein. After initiation of inhaled insulin therapy, all subjects returned to the BRU every week for the first month and every 2 weeks for the next 3 months. On each visit, fasting plasma glucose was measured, weight was recorded, and an interim medical history was obtained. HbA_1c_ and lipid profile were measured monthly. After 4 months all baseline studies were repeated. Exubera® (inhaled insulin) was not administered in the morning of the days on which the OGTT and lactate/tritiated glucose turnover studies were performed. Inhaled insulin was administered with breakfast, lunch, and dinner on the day of the 24-h glucose metabolic profile.

Nine of 11 subjects completed all portions of the study; 2 subjects failed to complete the end-of-study measurement of hepatic glucose production but completed the OGTT.

### 4.3. Analytical Procedures

Plasma glucose was measured in duplicate using the glucose oxidase method with a Beckman Glucose Analyzer II (Beckman, Fullerton, CA). HbA1c was determined by HPLC. Plasma insulin, C-peptide (Coat A Coat, Diagnostic Products, Los Angeles, CA), and glucagon (Double antibody, Siemens Health Care Diagnostics Inc, Los Angeles, CA) were determined by radioimmunoassay. Plasma lactate was determined by colorimetric method (Eton Bioscience Corp., San Diego, CA) and plasma FFA by standard colorimetric method (Wako Chemicals, Neuss, Germany). Plasma [3-^3^ H]glucose and [1-^14^C]glucose radioactivity levels were determined by the Somogyi procedure, as previously described [28]. Briefly, a plasma sample was deproteinized with barium hydroxide and zinc sulphate. The deproteinized supernatant fraction was passed through ion exchange column to separately elute fractions containing glucose and lactate and then evaporated to dryness to remove ^3^H2O, reconstituted with water, and radioactivity was counted.

### 4.4. Calculations and Statistical Analysis

Under steady state conditions following an overnight fast, the rate of basal HGP equals the rate of glucose uptake by all tissues in the body and was calculated as the tritiated glucose infusion rate (DPM/min) divided by the steady state plasma tritiated glucose specific activity (DPM/mg) during the last 30 min of tracer infusion [2]. Lactate turnover was calculated as the ^14^C-lactate infusion rate (DPM/min) divided by the steady state plasma lactate specific activity (DPM/mg). The percentage of glucose derived from lactate was calculated from the product-precursor relationship as follows: ^14^C-glucose specific activity divided by the lactate specific activity x 2 [17]. The rate of gluconeogenesis from lactate was calculated as the product of HGP and the percentage of glucose appearance derived from lactate.

The insulin secretion rate was calculated by deconvolution of the plasma C-peptide curve [29]. Insulin sensitivity was determined by the Matsuda index [30] as
$$\frac{10,000}{\sqrt{\left(\mathrm{FCP}\times \mathrm{FPG}\right)\cdot \left(\bar{\mathrm{CP}}\times \bar{G}\right)}}$$
where $\bar{\mathrm{CP}}$ and $\bar{G}$ represent the mean plasma C-peptide and glucose concentrations during the OGTT.

Insulin secretion was calculated as follows: ∆I_0-30_/∆G_0-30_ and ∆I_0-120_/∆G_0-120_. Beta cell function was calculated as the insulin secretion/insulin resistance (disposition) index = (∆I_0-120_/∆G_0-120_) X (Matsuda index of insulin sensitivity) [31,32]. Since following treatment with Inhaled insulin some patients had very high plasma insulin levels, the above indices were calculated using C-peptide in place of insulin. 

All data are presented as mean ± SEM. Within-group differences, i.e., pre- vs. post-Inhaled insulin treatment, were determined using the paired *t*-test or Wilcoxon test for non-normally distributed variables. Repeated measure ANOVA was used for the analysis of the 24-h profile. Correlation coefficients were calculated by least squares linear regression analysis. Results were considered statistically significant at *p* < 0.05.

## Figures and Tables

**Figure 1 ijms-20-04230-f001:**
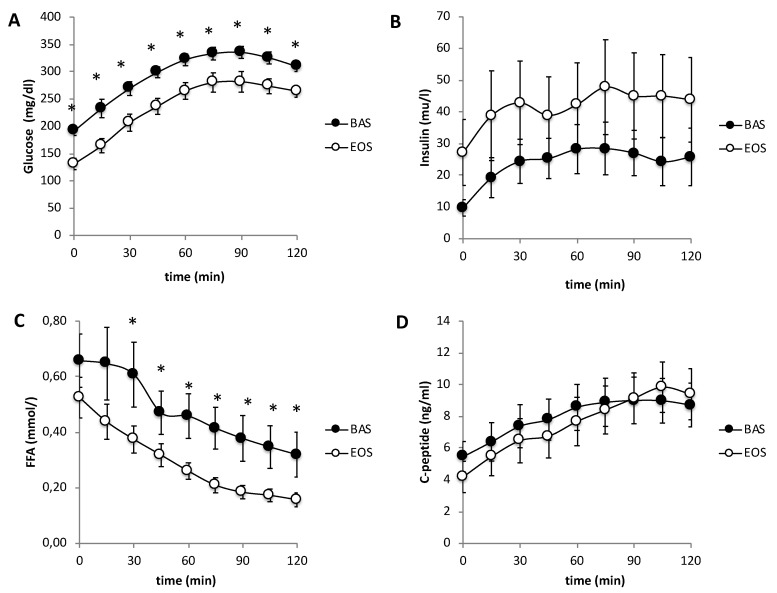
Plasma glucose (**A**), insulin (**B**), C-peptide (**C**), and FFA concentrations (**D**) during OGTT before (BAS) and after (EOS) 16 weeks of Inhaled insulin treatment. * *p* < 0.05.

**Figure 2 ijms-20-04230-f002:**
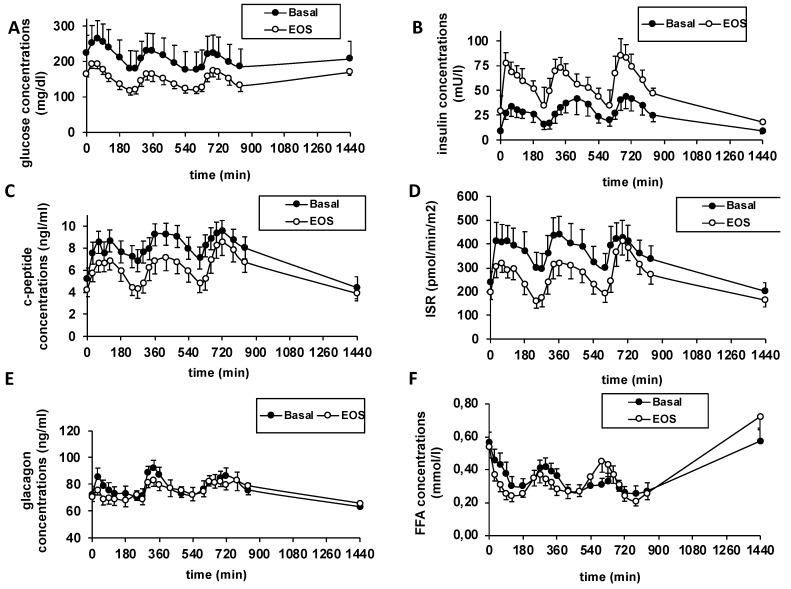
24-hr glucose profile before (Basal) and after (EOS) 16 weeks of treatment with Inhaled insulin. Glucose (**A**), insulin (**B**), C-peptide (**C**), ISR (insulin secretion rate, **D**), glucagon (**E**) and FFA (**F**) concentrations.

**Figure 3 ijms-20-04230-f003:**
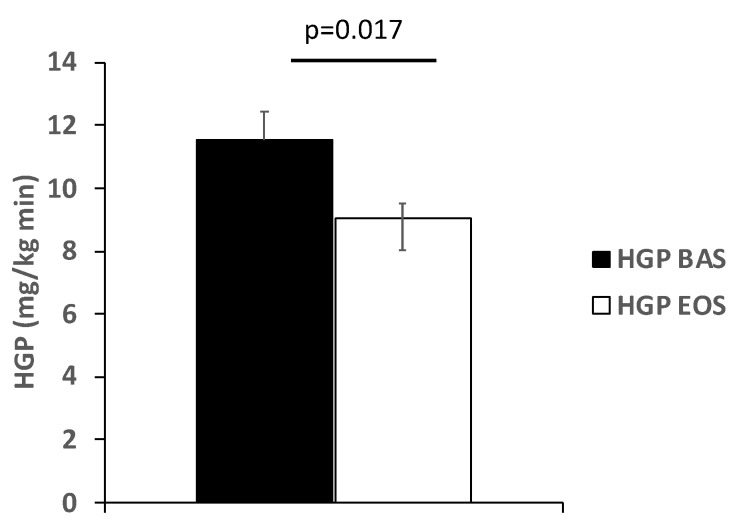
Fasting hepatic glucose production before (BAS) and after (EOS) 16 weeks of treatment with inhaled insulin.

**Table 1 ijms-20-04230-t001:** Clinical Characteristics of Study Subjects.

	Baseline	End of Study	*p*-Value
N =	11	11	
M/F	2/9	2/9	
Age (years)	53 ± 3	53 ± 3	ns
Weight (kg)	99.0 ± 4	101.3 ± 4.7	0.01
Fasting glucose (mg/dl)	215 ± 15	137 ± 11	<0.0002
2-h glucose mg/dl	309 ± 9	264 ± 11	*p* < 0.03
24-h glucose mg/dl	212 ± 17	141 ± 8	*p* < 0.0001
HbA1c (%)	10.3 ± 0.5	7.5 ± 0.3	<0.0001
Fasting Free fatty acids (µM/l)	665 ± 106	479 ± 61	0.05
Total Cholesterol (mg/dl)	177 ± 15	157 ± 11	ns
HDL Cholesterol (mg/dl)	35 ± 3	47 ± 9	<0.05
LDL Cholesterol (mg/dl)	105 ± 13	97 ± 11	ns
Triglycerides (mg/dl)	213 ± 39	120 ± 14	<0.05

**Table 2 ijms-20-04230-t002:** Correlation Analysis (Spearman Correlation Coefficients).

	ΔFPG	*p*-Value	ΔHbA1c	*p*-Value
ΔHGP	0.66	<0.05	0.45	0.22
ΔMatsuda index	0.65	<0.05	0.35	0.35
Δ Beta cell function	0.55	0.12	0.49	0.15

**Table 3 ijms-20-04230-t003:** Parameters of insulin secretion and insulin sensitivity.

	Baseline	End of Study	p-Value
***OGTT***			
ΔCP_0-30_/ΔG_0-30_	0.026 ± 0.008	0.031 ± 0.006	ns
ΔCP_0-120_/ΔG_0-120_	0.025 ± 0.005	0.031 ± 0.005	ns
Matsuda index	8.5 ± 1.8	16.4 ± 5.0	<0.05
Disposition index _0-30_	0.14 ± 0.05	0.30 ± 0.03	<0.02
Disposition index _0-120_	0.16 ± 0.02	0.33 ± 0.06	<0.02
***24 h***			
Mean glucose (mg/dl)	212 ± 17	141 ± 11	0.01
Mean insulin (mg/dl)	25 ± 7	41 ± 9	0.009
Mean C-peptide (pg/ml)	7.3 ± 1.2	5.9 ± 0.8	ns
Mean glucagon (ng/ml)	78 ± 5	76 ± 5	ns
Mean FFA (µmol/l)	214 ± 26	183 ± 21	ns

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
