# Peer review of "Mechanism of Action of Inhaled Insulin on Whole Body Glucose Metabolism in Subjects with Type 2 Diabetes Mellitus"

_ijms, 2019, doi:10.3390/ijms20174230_

Round 1

Reviewer 1 Report

The paper “Mechanism of Action of Inhaled Insulin on whole body glucose metabolism in subjects with Type 2 diabetes” describes the action of inhaled insulin in 11 patients.

This paper possibly has the worst abstract I have ever read. It has 10 abbreviations, most are not defined, and some of them even do not match within the abstract (e.g. A1c and HbA1c). According to the journals “instructions for authors” it should not have subheadings, which it has, and it is too long according to these same instructions. It uses uM, I assume this should be µM? (This use continues intermittently in the rest of the manuscript). In addition to this, it list every single measurement, means and SD, done in the study, this makes it very difficult to read. Please rewrite the abstract so that it lists background, main findings, and how they relate to your conclusions, and define abbreviations!

The rest of the paper is an improvement compared to the abstract, however the sloppy writing continues.

General comments:  Exubera is, as you say, no longer on the market. I am missing some link between Exubera and how it can relate to other inhaled insulin products. Are these results directly translatable to Afrezza? How similar are Exubera and Afrezza, and does this paper support inhaled insulin in general or only a product not obtainable by the public anymore?

There has been some concern that inhaled insulin may be damaging to the lungs. Did you do any lung measurements other than FEV1?

The patient population has improved control over their diabetes after the trial, however if I understand correctly they all followed a meal plan laid out by a dietitian during the whole trial. How does this presumably large change in their diet affect your results? Please add a discussion on this.

However convincing the result may be, the population is very small as you yourself comment upon. Together with the fact that the patients were on a controlled diet, the statement “inhaled insulin may prove to be a panacea for the insulin requiring type 2 diabetic patient” is a bit too optimistic.  

The exact same data is found both in the table and listed in the text. This is not necessary, use the text to elaborate on the data, or delete the table.

Specific comments:

Abbreviations: these are now defined, but not the first time they appear, e.g. T2D, coincidentally this is abbreviated T2DM in the abstract. Some are not defined at all, e.g. FEV1, NGT and CRC?

Figures: Figure 2 comes before figure 1?

The text in the figures is illegible. They are labeled “before EOS” or “EOS”, what is EOS?

Statistics:

Body weight in table 1 has a p-value of <0.02, in the text it is given as p=0.01?

What statistical program and version did you use?

In the section “Insulin secretion and insulin sensitivity”, you state “p=ns”, I assume “not significant”? Please state the actual p value, or say “the results were not significant”.

Methods: the section “Analytical procedures” is very short, please add some references for the analysis made.

Text: Inhaled insulin is capitalized sometimes and not others in the middle of sentences please be consistent.

 It may be a problem with the pdf, but some words appear in a larger font size, e.g. insulin on page 9, line 45, 50, 51 and 52.

On page 8, line 10 you write: “this was felt to be unrelated to inhaled insulin treatment”, please use a different word than “felt”!

Author Response

Reply to REVIEWER 1

We thank the reviewer for the useful comments

This paper possibly has the worst abstract I have ever read. It has 10 abbreviations, most are not defined, and some of them even do not match within the abstract (e.g. A1c and HbA1c). According to the journals “instructions for authors” it should not have subheadings, which it has, and it is too long according to these same instructions. It uses uM, I assume this should be μM? (This use continues intermittently in the rest of the manuscript).

In addition to this, it list every single Measurement, means and SD, done in the study, this makes it very difficult to read. Please rewrite the abstract so that it lists background, main findings, and how they relate to your conclusions, and define abbreviations! The rest of the paper is an improvement compared to the abstract, however the sloppy writing continues.

Answer: We thank the reviewer for his/her comments. The abstract has been shortened, abbreviations described and uM, corrected as μM

General comments: Exubera is, as you say, no longer on the market. I am missing some link between Exubera and how it can relate to other inhaled insulin products. Are these results directly  Translatable to Afrezza? How similar are Exubera and Afrezza, and does this paper support inhaled insulin in general or only a product not obtainable by the public anymore?

There has been some concern that inhaled insulin may be damaging to the lungs. Did you do any lung measurements other than FEV1?

Answer: Exubera and Afrezza were very similar as type of insulin, but very different as far as delivery system: Exubera’s delivery system was large,  and dosed in milligrams whereas Afrezza’s delivery system is  much smaller, and dosed in units and provides a simple dosing conversion chart. The modifications implemented with Afrezza resulted in  a more user friendly  device and thus  compliance rate  is much higher than that of  Exubera. Yes the authors support the use of the inhaled insulin route.

Patients were excluded if they had pre-existing  chronic lung disease i.e, Asthma, COPD, but besides FEV1 no other measurements were done.

The patient population has improved control over their diabetes after the trial, however if I understand correctly they all followed a meal plan laid out by a dietitian during the whole trial. How does this presumably large change in their diet affect your results? Please add a discussion on this.

Answer: As we have written in the manuscript in page 3 “over the subsequent 4 months, subjects continued to consume a weight maintaining diet containing 50% carbohydrate, 30% fat, and 20% protein.” The diet was very similar to what the patients were consuming before the trial, but we wanted to avoid excessive weight gain due to insulin therapy.

We have also discussed the implications of this in the discussion at page 8 reporting that “Diet and exercise can influence whole body glucose homeostasis; hence subjects were asked to maintain a similar weight maintaining diet and exercise regimen throughout the study duration. No weight loss was observed in any of the subjects and in fact there was a mean weight gain in the group. Hence we believe that diet and exercise likely did not contribute to  the improvement in  glycemic control at beginning and end of study.”

However convincing the result may be, the population is very is very small as you yourself comment upon. Together with the fact that the patients were on a controlled diet, the statement “inhaled insulin may prove to be a panacea for the insulin requiring type 2 diabetic patient” is a bit too optimistic.

Answer:   we have rephrased the sentence that now reads as “… due to its beneficial effects on whole body glucose homeostasis, inhaled insulin may prove to be an easier  option and may help reduce clinical inertia for the insulin requiring type 2 diabetic patient.”

Figures: Figure 2 comes before figure 1? The text in the figures is illegible. They are labeled “before EOS” or “EOS”, what is EOS?

Answer: EOS is end of study, we have now added a list of abbreviations in the text

The exact same data is found both in the table and listed in the text. This is not necessary, use the text to elaborate on the data, or delete the table.

Answer: We have left the data in the tables, and added new tables 2 and 3.

Specific comments:

Abbreviations: these are now defined, but not the first time they appear, e.g. T2D, coincidentally this is abbreviated T2DM in the abstract. Some are not defined at all, e.g. FEV1, NGT and CRC?

Answer: The word panacea has been deleted. Figures are appropriately labelled. “EOS=End of study” is now in the legend and text. Text data has been modified. Table is kept. Abbreviations clarified.

Statistics:

Body weight in table 1 has a p-value of <0.02, in the text it is given as p=0.01? What statistical program and version did you use? In the section “Insulin secretion and insulin sensitivity”, you state “p=ns”, I assume “not significant”? Please state the actual p value, or say “the results were not significant”.

Answer: This has been corrected. We used Statview (SAS) as statistical program

Methods: the section “Analytical procedures” is very short, please add some references for the analysis made. Text: Inhaled insulin is capitalized sometimes and not others in the middle of sentences please be consistent. It may be a problem with the pdf, but some words appear in a larger font size, e.g. insulin on page 9, line 45, 50, 51 and 52.

Answer: Analytical procedure session is expanded and references are added.

On page 8, line 10 you write: “this was felt to be unrelated to inhaled insulin treatment”, please use a different word than “felt”!

Answer: This has been modified.

Reviewer 2 Report

The manuscript describes a study investigating the mechanisms of action of inhaled insulin on glucose metabolism in subjects with Type 2 diabetes (T2D).

In general, the study addresses a relevant and interesting topic related to alternative routes of insulin administration for subjects with T2D. The manuscript is well written, readable and the scope is easy to follow. However, there are some major limitations and concerns related to study design and statistical analyses and they are listed below.

1. Please, clarify why you did not include a control group.

2. As the study did not include a control group, the results may not be linked to the inhaled insulin but to other confounding factors or both. Please identify possible confounders and address the issue in the discussion part.

3. The statistical analyses are insufficient. Instead of using paired t-test, the authors should consider to use mixed models in order to adjust for covariates such as individual weight change, dietary intake of nutrients known to affect blood glucose, physical activity etc.  

4. The authors have not included data on dietary intake and physical activity. There are several nutrients that may affect whole body glucose metabolism, such as fiber, fat quantity/quality and sugar, and alcohol. Also physical activity affects glycemic control. It is well known that subjects participating in interventions unconsciously improve their diet and are more physical active. If possible, please add data on dietary intake during the study and use these data as covariates in statistical analyses as described above.

5. Figure 1 and 2 are of very bad quality and new figures of better quality should be included. Please also explain abbreviations in the figure text. The figure text should include more details of the data presented. Title on the x-axis in figure 1 are missing. Why are figure 2 placed before figure 1?

6. Data for some variables presented in the text and included in the statistical analyses are not presented in the figures or table 1. Please include all the variables that are presented in the text. This includes the correlation data.

6.  Start the discussion with a very short summary of the results.

7. Make sure that abbreviations are explained first time, also in the abstract.

Author Response

Reply to REVIEWER 2

We thank the reviewer for the useful comments

The manuscript describes a study investigating the mechanisms of action of inhaled insulin on glucose metabolism in subjects with Type 2 diabetes (T2D). In general, the study addresses a relevant and interesting topic related to alternative routes of insulin administration for subjects with T2D. The manuscript is well written, readable and the scope is easy to follow. However, there are some major limitations and concerns related to study design and statistical analyses and they are listed below.

Please, clarify why you did not include a control group. As the study did not include a control group, the results may not be linked to the inhaled insulin but to other confounding factors or both. Please identify possible confounders and address the issue in the discussion part,

Answer: In the current study we did not include a control group as it was primarily designed to explain mechanisms whereby a short acting inhaled insulin can also reduce fasting plasma glucose. However, data on fasting HGP were compared with previously published data from our group (see paragraph on HGP at page 6).

Diet and exercise can influence whole body glucose homeostasis; hence subjects were asked to maintain a similar weight maintaining diet and exercise regimen throughout the study duration. No weight loss was observed in any of the subjects and in fact there was a mean weight gain in the group. Hence we believe that diet and exercise likely do not contribute to  the glycemic benefits seen at beginning and end of study.

Several trials (phase II and III) have shown that exubera is similar or superior to sc insulin in reducing HbA1c in T2D patients. The study by Hollander showed similar weight gain.  Patients were randomized to 6 months’ treatment with either premeal inhaled insulin plus a bedtime dose of Ultralente  or at least two daily injections of subcutaneous insulin . HbA1c decreased similarly in the inhaled (0.7%) and subcutaneous (0.6%) insulin groups. HbA1c < 7.0% was achieved in more patients receiving inhaled (46.9%) than subcutaneous (31.7%)insulin. Overall hypoglycemia (events per subject-month) was slightly lower in the inhaled (1.4 events) than in the subcutaneous (1.6 events) insulin group, with no difference in severe events. Other adverse events, with the exception of increased cough in the inhaled insulin group, were similar. No difference in pulmonary function testing was seen.

The statistical analyses are insufficient. Instead of using paired t-test, the authors should consider to use mixed models in order to adjust for covariates such as individual weight change, dietary intake of nutrients known to affect blood glucose, physical activity etc.

Answer: We thank the reviewer of the comment and we agree. However the limited number of the subjects and the lack of a control group does not allow to use mixed models.

The authors have not included data on dietary intake and physical activity. There are several nutrients that may affect whole body glucose metabolism, such as fiber, fat quantity/quality and sugar, and alcohol. Also physical activity affects glycemic control. It is well known that subjects participating in interventions unconsciously improve their diet and are more physical active. If possible, please add data on dietary intake during the study and use these data as covariates in statistical analyses as described above.

Answer:  As we have written in the paper at page 3 “..over the subsequent 4 months, subjects continued to consume a weight maintaining diet containing 50% carbohydrate, 30% fat, and 20% protein.” The diet was very similar to what the patients were consuming before the trial, but we wanted to avoid excessive weight gain due to insulin therapy. We did not record data on physical activity. However, as previously stated, no weight loss was observed in any of the subjects and in fact there was a mean weight gain in the group. Hence we believe that diet and exercise likely do not contribute to any of the glycemic benefits we see at beginning and end of study.

Figure 1 and 2 are of very bad quality and new figures of better quality should be included. Please also explain abbreviations in the figure text. The figure text should include more details of the data presented. Title on the x-axis in figure 1 are missing. Why are figure 2 placed before figure 1?

Answer: This has been modified and edited.

Data for some variables presented in the text and included in the statistical analyses are not presented in the figures or table 1. Please include all the variables that are presented in the text. This includes the correlation data.

Answer: This is done and Table 2 is inserted showing correlation data.

Start the discussion with a very short summary of the results.

Answer: This is added.

Make sure that abbreviations are explained first time, also in the abstract.

Answer: This was done.

Reviewer 3 Report

Comments and Suggestions for Authors

This research article addresses effects of inhaled insulin on endogenous glucose production (HGP), plasma glucose and free fatty acid concentrations. This is very well written, and the objective of conducting this research is persuasive.

However, several limitations of this study are noted.

Several suggestions are as follows;

-In introduction, it would be helpful to add the explanation of Exubera insulin inhalation system (device).

-In introduction, it would be helpful to address the some of the pharmacodynamics and pharmacokinetics profile of inhaled insulins.

-Power calculation should be mentioned.

Author Response

Reply to REVIEWER 3

We thank the reviewer for the useful comments

This research article addresses effects of inhaled insulin on endogenous glucose production (HGP), plasma glucose and free fatty acid concentrations. This is very well written, and the objective of conducting this research is persuasive.

However, several limitations of this study are noted.

Several suggestions are as follows;

-In introduction, it would be helpful to add the explanation of Exubera insulin inhalation system (device). -In introduction, it would be helpful to address the some of the pharmacodynamics and pharmacokinetics profile of inhaled insulins.

Answer: Description of inhaled insulin device, pharmacokinetic data of inhaled insulin is added.

-Power calculation should be mentioned.

Answer: Based upon previously published results from our lab, we anticipate that basal hepatic glucose production (HGP) will be elevated in type 2 diabetic patients (12.2±3.9μmol/kg.min) (mean±SD) and will decline to 10.0±3.3 μmol/kg.min following 4 months of Exubera therapy. In order to observe a reduction in HGP of 2.2±1.7 μmol/kg.min with an alpha level of 0.05 and a beta effect of 0.80, we would require ~15 type 2 diabetic patients. Although the trial was stopped interim and could only complete 11 patients, Figure 3 shows that the reduction in HGP was significant.

Thus, we believe that our data are of interest and could support the design of future studies on new inhaled insulin compounds that are on the pipeline

Reviewer 4 Report

The manuscript entitled “Mechanism of action of inhaled insulin on whole body glucose metabolism in subjects with type 2 diabetes” (Mehta et al.) refers to Exubera action for patients with type 2 diabetes.

Minor revision:

The authors tested short-acting insulin Exubera on 11 poorly controlled T2D patients. However, what is poorly controlled? Please, define more concrete.

In abstract part (that should stay as a single paragraph without headings) there are no units after the parameters in brackets and many abbreviations without previous explanation.

In the manuscript, there are many abbreviations that should be explained when mentioned firstly.

The style of writing should be uniform, f. e. writing of times 0800 and 10-12h

Major revision:

The authors mention in the conclusion that the number of patients enrolled in the study is too small. I recommend to add more “poorly controlled” patients and also patients with previous insulin treatment.

Author Response

Reply to REVIEWER 4

We thank the reviewer for the useful comments

The manuscript entitled “Mechanism of action of inhaled insulin on whole body glucose metabolism in subjects with type 2 diabetes” (Mehta et al.) refers to Exubera action for patients with type 2 diabetes.

Minor revision:

The authors tested short-acting insulin Exubera on 11 poorly controlled T2D patients. However, what is poorly controlled? Please, define more concrete.

In abstract part (that should stay as a single paragraph without headings) there are no units after the parameters in brackets and many abbreviations without previous explanation.

In the manuscript, there are many abbreviations that should be explained when mentioned firstly.

The style of writing should be uniform, f. e. writing of times 0800 and 10-12h

Answer:  Poorly controlled is now defined.

Abstract is rectified as well as all other suggestions are made for more uniformity of manuscript.

Major revision:

The authors mention in the conclusion that the number of patients enrolled in the study is too small. I recommend to add more “poorly controlled” patients and also patients with previous insulin treatment.

Answer: This study was completed a while ago and inhaled insulin exubera is no longer commercially available  and thus it  will not be feasible to  add more patients to the study.

Round 2

Reviewer 2 Report

Thanks for the revised version. Although the quality of the paper has increased it still needs to be improved.

There are still abbreviations in the abstract that are not spelled out (line 16, FPG)

Given the fact that there are no control group in the present study, I would recommend that the authors modify the conclusion in the abstract and in the discussion. With the current design it is not possible to draw the conclusion that the improvement in glycemic control is due to inhaled insulin. So please rephrase using words such as “may indicate” and state that further study needs to be performed.

Author Response

We have rephrased the conclusion that now states

“The present study has several limitations, i.e., the relatively small number of subjects and the lack of a group of subjects treated with conventional insulin therapy for comparison with Inhaled insulin, as it was designed to understand the mechanisms of action of inhaled insulin in improving metabolic control in type 2 diabetic patients already demonstrated in previous studies [25, 26, 32].

In summary, our study shows that thrice daily inhaled insulin administration effectively reduced the fasting and postprandial plasma glucose concentrations and markedly improved glycemic control (DHbA1c = -2.8%) in poorly controlled T2D patients by augmenting tissue sensitivity to insulin and enhancing beta cell function.Inhaled insulin may be an advantage in poorly controlled diabetic patients who are averse to taking insulin injections, with faster onset of action and a shorter duration thereby reducing risk of hypoglycemia. Moreover, due to its beneficial effects on whole body glucose homeostasis, inhaled insulin may prove to be an easier option and may help reduce clinical inertia for the insulin requiring type 2 diabetic patients.”

Reviewer 3 Report

Sample size is too small.

Additional experiments are needed.

Author Response

We believe our study provides new evidences on the mechanism of action on inhaled insulin

As previously stated it is not possible to add new subjects nor additional experiments since Exubera is no longer on the market. However, since as stated at page 2  “There is renewed interest in alternative routes of insulin delivery in order to improve patient adherence to therapy as well as improve quality of life. This novel delivery route may also combat physician and patient inertia in intensifying treatment when needed.”

Limitation of the study are clearly stated at the end of discussion

Reviewer 4 Report

The manuscript is already better written in compare with how it was. However, there are many shortcomings. The line spacing is not uniform throughout the manuscript. Also, the abbreviations AM, PM are not uniform throughout the paper (7AM vs 7 AM, 7.30 vs 7:30). Throughout the text, there are too many double space-bars used (f. e. line 28 before ..Fasting, lines 30, 32 and many many others. The abbreviation HbA1c is not specified. After withdrawal of Exubera, there is a need to claerly state why this study is relevant.

Author Response

We have deleted the double spaces, corrected the abbreviations and defined HbA1c (glycated hemoglobin)

After withdrawal of Exubera, there is a need to claerly state why this study is relevant.

This is already stated in the introduction (page 1 line 35)

“Although Exubera® was initially approved in 2006 by FDA [12], it was withdrawn from the US in 2007 for poor patient acceptance and thus lack of commercial viability. However, another inhaled insulin Afrezza®, which is similar to Exubera®, has been approved by FDA and is currently available [13]. There is renewed interest in alternative routes of insulin delivery in order to improve patient adherence to therapy as well as improve quality of life. This novel delivery route may also combat physician and patient inertia in intensifying treatment when needed.”